# Identification of Predictive Factors for Overall Survival and Response during Hypomethylating Treatment in Very Elderly (≥75 Years) Acute Myeloid Leukemia Patients: A Multicenter Real-Life Experience

**DOI:** 10.3390/cancers14194897

**Published:** 2022-10-06

**Authors:** Matteo Molica, Carla Mazzone, Pasquale Niscola, Ida Carmosino, Ambra Di Veroli, Cinzia De Gregoris, Fabrizio Bonanni, Salvatore Perrone, Natalia Cenfra, Luana Fianchi, Anna Lina Piccioni, Antonio Spadea, Giovanni Luzi, Andrea Mengarelli, Laura Cudillo, Luca Maurillo, Livio Pagano, Massimo Breccia, Luigi Rigacci, Paolo De Fabritiis

**Affiliations:** 1Hematology and Stem Cell Transplant Unit, St. Eugenio Hospital, 00144 Rome, Italy; 2Hematology, Department of Translational and Precision Medicine, Policlinico Umberto I, Sapienza University, 00185 Rome, Italy; 3Hematology Division, Ospedale Belcolle, 01100 Viterbo, Italy; 4Hematology, Department of Biomedicine and Prevention, University Tor Vergata, 00133 Rome, Italy; 5Hematology, Polo Universitario Pontino, S.M. Goretti Hospital, 04100 Latina, Italy; 6Institute of Hematology, Fondazione Policlinico Universitario A. Gemelli IRCCS, 00168 Rome, Italy; 7Hematology and Stem Cell Transplant, San Giovanni Addolorata, 00184 Rome, Italy; 8Hematology and Stem Cell Transplant Unit, IRCCS Regina Elena National Cancer Institute, 00144 Rome, Italy; 9Hematology and Stem Cell Transplant, San Camillo Forlanini Hospital, 00152 Rome, Italy

**Keywords:** acute myeloid leukemia, very elderly, real-life experience

## Abstract

**Simple Summary:**

Intensive induction strategies are rarely used for older patients in community on-cology practice, with comorbidities being the major cause of contraindication. We conducted a multicentric retrospective study to evaluate activity and safety in a real-life setting of hypomethylating drugs (HMAs) in patients older than 75 years with AML. In multivariate analysis, age (≥80), Charlson comorbidity index (≥3), creatinine clearance and the type of best response (≥PR) during treatment maintained independent significance in predicting survival. Furthermore, our data show that HMAs have similar efficacy compared to pivotal trials and are well tolerated in a setting of very elderly patients with several co-comorbidities.

**Abstract:**

Elderly patients represent the most challenging and hard-to-treat patient population due to dismal characteristics of the disease, such as secondary-acute myeloid leukemia (AML), enrichment of unfavorable molecular genes (*TP53*) and comorbidities. We conducted a multicentric retrospective study to evaluate activity and safety in a real-life setting of hypomethylating drugs (HMAs) in patients older than 75 years with AML. Between September 2010 and December 2021, 220 patients were treated, 164 (74.5%) received AZAcitidine and 56 DECitabine; most patients (57.8%), received more than four cycles of HMAs. The best response obtained was CR in 51 patients (23.2%), PR in 23 (10.5%) and SD in 45 (20.5%); overall transfusion independence was obtained in 47 patients (34%), after a median of 3.5 months. The median OS (mOs) was 8 months (95% CI 5.9–10.2), with 1- and 2-years OS of 39.4% (95% CI 32.7–46) and 17.4% (95% CI 11.7–23.1), respectively; similar mOS was observed according to HMA treatment (AZA 8.3 vs. DEC 7.8 months, *p* = 0.810). A subset of 57 long survivors (44 in AZA group and 13 in DEC group) received at least 12 cycles of HMAs, their mOS was 24.3 months. In multivariate analysis, age (≥80), Charlson comorbidity index (≥3), creatinine clearance and the type of best response (≥PR) during treatment maintained independent significance in predicting survival. Infectious complications, most frequently pneumonia (35) and septic shock (12), were lethal in 49 patients (22.2%). Our data show that HMAs have similar efficacy compared to pivotal trials and are well tolerated in a setting of very elderly patients with several co-comorbidities.

## 1. Introduction

Acute myeloid leukemia (AML) is primarily a disease in older adults, with a median age of 68 years at diagnosis [1]. Analysis of a Surveillance, Epidemiology and End Results (SEER) database shows that approximately one-third of patients are older than 75 years [2]. Very elderly patients (>75 years old) with AML have a greater prevalence of comorbidity and inferior performance status, both of which augment the risk of toxicity and early death with intensive treatment or new target therapies. Furthermore, age is associated with adverse molecular, cytogenetic, and biological features that confer chemo-resistance and predict inferior outcomes. In addition, in this subset, there is a higher incidence of secondary AML, including AML with preceding myelodysplastic syndrome and therapy-related AML, and of multidrug resistance phenotypes [3]. Therefore, the estimated 2-year survival rate of patients aged ≥ 65 accounts for less than 20% [4]. AML in older patients is an escalating clinical problem because our population is aging and the decision-making treatment remains an arduous challenge for physicians, especially in a very elderly setting in which supportive care often represents the only tolerable therapeutic approach. 

The availability of hypomethylating agents (HMAs) after 2004 represented an important expansion in treatment options for older patients with AML. Large randomized clinical studies have confirmed the clinical benefit of HMAs for AML patients (complete remission and transfusion independence rates ranging from 15% to 20% and 30% to 40%, respectively) but have not demonstrated strongly consistent improvements in survival [5,6,7,8]. The toxicity and mortality associated with intensive chemotherapy and the lack of alternative non-intensive treatments for older AML patients contributed to the adoption of HMAs as the de facto standard of care for this setting [9,10,11]. Real-world evidence for the clinical benefits of HMAs in much older AML patients (>75 years old) is very limited, and the two approved HMAs have not been directly compared in this setting. This gap in evidence continues to represent a crucial issue, as HMAs have become the backbone for combination regimens (e.g., with venetoclax), with approval based on single-arm studies without an HMA monotherapy control arm [12]. This study aims to assess the real benefit of HMAs in a very elderly population of patients with AML, comparing the two HMA agents, investigating clinical and biological factors potentially affecting response and outcomes and questioning, according to toxicities, whether HMAs monotherapy is enough in this arduous treatable subset. 

## 2. Patients and Methods

Two-hundred-twenty consecutive very elderly AML patients (≥75 years old) who received HMAs as first-line treatment outside of clinical trials at eight different hematologic centers in the Lazio region (Italy) between September 2010 and December 2021 were analyzed. One-hundred-sixty-four (74.5%) patients received subcutaneous AZA 75 mg/m^2^ for 7 days according to the 5 + 2 schedule every 4 weeks and 56 (25.5%) patients received intravenous DEC 20 mg/m^2^ for 5 consecutive days every 4 weeks until disease progression or unacceptable toxicity. Patients who had received prior HMAs or other chemotherapy approaches were excluded from the analysis. The diagnosis of AML was performed according to the World Health Organization (WHO) 2016 criteria [3]. Data collected included blood count values, bone marrow blast count, biological and cytogenetic characteristics at the time of AML diagnosis. The prognostic risk was assessed according to the 2017 European Leukemia Net (ELN) prognostic classification [4]. The database also included patient-related characteristics at baseline such as age, comorbidity, body mass index (BMI), renal function, the presence of infection (pneumonitis, sepsis or others) and transfusion dependence at baseline. The Charlson comorbidity index (CCI) was used as an indicator of comorbidity [13]. Comorbidities were evaluated at baseline, before HMAs treatment. All information regarding concomitant diseases and drug usage was recorded in each case history and thereafter used for this retrospective evaluation. The CCI is a list of 19 comorbid conditions: each condition has a weight assigned from 1 to 6, which is derived from the relative risk estimates of a proportional hazard regression model using clinical data. The estimated glomerular filtration rate (eGFR) was calculated through the Chronic Kidney Disease Epidemiology Collaboration (CKD-EPI) equation [14] for all patients. The CKD-EPI equation is: GFR = 141 × min(Scr/κ,1)^α^ × max(Scr/κ, 1)^−1.209^ × 0.993^Age^ × 1.018 [if female] × 1.159 [if black], 
where Scr is serum creatinine (mg/dL), κ is 0.7 for females and 0.9 for males, α is −0.329 for females and −0.411 for males, min indicates the minimum of Scr/κ or 1, and max indicates the maximum of Scr/κ or 1. BMI was calculated at the time of the start of HMA treatment. BMI was defined as the individual’s body weight in kilograms divided by the square of his/her height, which produces a unit of measure of kg/m^2^. The response to treatment was evaluated according to the revised criteria established by the International Working Group (IWG) of AML [15]. The study was approved by the Institutional Review Board of each institution and was conducted according to the Helsinki declaration.

Categorical data were expressed as counts and percentages, whereas continuous variables were reported as medians with 25th to 75th interquartile range (IQR). Differences in the study groups concerning characteristics and treatment responses were estimated using the chi-square test or the Fisher exact test for categorical covariates and the Mann–Whitney U test for continuous variables. Overall survival (OS) was calculated from the start of therapy to death from any cause or the date of the last follow-up. Event-free survival (EFS) was calculated from the start of therapy to the date of the end of treatment caused by the progression of the disease (PD) or severe complications or death. Probabilities of OS and EFS were estimated using the Kaplan–Meier analysis and compared using the log-rank test. Univariate and multivariate Cox proportional hazards and logistic regression models assessed the association of covariates with survival and response rates, respectively. All *p* values < 0.05 were considered statistically significant. All statistical analyses were performed using IBM SPSS Statistics, version 27.

## 3. Results

### 3.1. Patients’ Characteristics

Baseline patients’ characteristics are summarized in Table 1. The median age was 78.2 years (IQR 75–86.2 years). Of the patients, 135 (61.4%) had de novo AML and 85 (38.6%) had secondary AML (s-AML). Among the s-AML group, 18 (8.2%) patients had a prior solid tumor (all of them had received chemotherapy or radiotherapy), while 47 (21.4%), 18 (8.2%) and 6 (0.8%) presented a previously documented myelodysplastic syndrome (none of them had received HMAs), a Philadelphia negative myeloproliferative disorder (all of them had received hydroxyurea treatment) and a non-Hodgkin’s lymphoma (all of them had received chemotherapy), respectively. The majority of patients (49.1%) had a CCI ≥ 3, with 88 patients (39.5%) who presented an ECOG ≥ 2. More than half of the patients (55.9%) showed an increased BMI (≥25) and 37 (16.8%) patients had a decreased eGFR (≤60 mL/min) at baseline. There were 49 (22.9%) patients who presented a documented infection at the time of diagnosis (12.7% pneumonia, 3.2% sepsis and 6.4% others), while 137 (62.7%) had a transfusion requirement of blood cells and/or platelets. Considering bone marrow blasts at baseline, 78 patients (35.5%) had blasts between 20–30%, 74 (33.6%) between 30–50% and 68 (30.9%) more than 50%. Among 198 patients with evaluable cytogenetic profiles, 86 (43.5%) had normal and 42 (21.3%) had complex karyotype (CK); 18 (9%) patients were chromosome 5 deletion carriers, while another 13 (6.5%) had monosomal karyotype (MK) for chromosome 7; and 15 (7.5%) patients harbored a trisomy of chromosome 8 and 24 (12.2%) patients had different alterations. A genetic risk assessment, according to the ELN recommendation, has been possible for 205 patients; among them, 17 (8.3%), 100 (48.8%) and 88 (42.9%) patients were stratified as low-, intermediate- and high-risk, respectively. Comparing the two HMA groups, patients treated with AZA were older than those treated with DEC (median age 78 years (IQR 76–82) vs. 77 (IQR 76–78), respectively, *p* < 0.001). The median WBC count at diagnosis was significantly higher for the DEC group (5.65 × 10^9^/L (IQR 2.48–13.87) vs. AZA 2.62 (IQR 1.32–7.93), *p* = 0.002). No differences regarding the other disease characteristics and demographic data were observed among the two treatment groups. The median follow-up was 7.8 months (IQR 3.2–16.2) for the entire cohort.

### 3.2. Response to Treatment

The median number of HMA cycles was 5 (IQR 1–46), with 93 patients (42.2%) who received less than 4 cycles. Of the total of 220 patients, 126 (57.3%) received at least 4 cycles of treatment and only 8 patients (3.6%) received HMAs at reduced doses. Ninety-four patients discontinued treatment within 4 cycles due to progressive disease (11.3%), uncontrollable toxicities (16.3%) and death (15%). At the 4th cycle of HMA, complete remission (CR) was reached in 41 (18.6%), partial remission (PR) in 23 (10.5%), stable disease (SD) in 46 (20.9%) patients, while 16 had progressive disease (PD) (7.6%.). There were 79 patients (35.9%) who received 8 or more cycles and for 65 of them, bone marrow was evaluated. Of these, 46 had <30% and 19 ≥30% marrow blasts. Thus, at the 8th cycle of HMA, CR was reached in 36 (16.4%), PR in 7 (3.2%), SD in 19 (8.6%) patients, while 4 had PD (1.8%). Among 138 patients with known transfusion dependence at diagnosis, overall transfusion independence (TI) was obtained in 47 patients (34%), after a median time of 3.5 months (IC 2.7–4.2 months).

During treatment with HMA, the best response obtained was CR in 51 (23.2%), PR in 23 (10.5%) and SD in 45 (20.5%) patients. Comparing the response to treatment in the two groups, no differences were found in CR rate between AZA (34/164 pts, 20.7%) and DEC (17/56, 30%) (*p* = 0.147) and in time to CR (4.6 vs. 3.9 months; *p* = 0.112).

Results of univariate logistic regression model analysis for CR are provided in Appendix A.

In multivariate logistic regression analysis, variables that were significantly associated with higher odds of achieving CR included ECOG < 2 [OR 2.96 (95% CI 1.13–7.74), *p* = 0.027], CCI < 3 [OR 4.47 (1.70–11.71), *p* < 0.001], whereas a poor predictor of response remained s-AML type [OR 0.28 (95% CI 0.10–0.78), *p* = 0.015] and bone marrow blasts ≥ 50% at diagnosis [OR 0.12 (95% CI 0.03–0.44), *p* = 0.001)].

### 3.3. Overall Survival

The median OS (mOs) was 8 months (95% CI 5.9–10.2), with 1- and 2-years OS of 39.4% (95% CI 32.7–46) and 17.4% (95% CI 11.7–23.1), respectively (Figure 1). Results of the Kaplan–Meier overall survival analysis are shown in Table 2.

No difference in OS was observed according to HMA treatment [AZA 8.3 vs. DEC 7.8 months, *p* = 0.810] (Figure 2). Time from diagnosis to start of therapy (TDT) (<15 vs. >15–30 vs. >30 days) did not affect mOS [7.5 (95% CI 4.5–10.5) vs. 11 (95% CI 4.5–17.4) vs. 7.7 months (95% CI 4.8–10.6), respectively; *p* = 0.399] (Figure 3). The following features were associated with improved mOS: age < 80 years [10.5 months (95% CI 7.4–13.5) vs. 6.2 months (95% CI 3.0–9.5) in patients ≥80 years; *p* = 0.001], ECOG <2 [9.8 months (95% CI 7.5–12) vs. 2.3 months (95% CI 1.3–3.8) for ECOG ≥ 2; *p* < 0.001], CCI < 3 [14.3 months (95% CI 7.9–20.7) vs. CCI ≥ 3 6.7 (95% CI 5–1–8.2), *p* = 0.029], and the achievement of TI during HMAs [19.3 months (95% CI 15.4–27.3) vs. 3.3 months (95% CI 0.8–5.8) in patients who continued requiring transfusions during HMAs; *p* < 0.001].

Significant differences were observed between patients with bone marrow blasts between 20 and 29% [13.7 months (95% CI 9.8–17.6)] and patients with bone marrow blasts ≥50% [4.1 months (95% CI 1.7–6.5); *p* < 0.001], but no difference was documented between patients with bone marrow blasts between 20 and 29% and patients with bone marrow blasts between 30 and 50% [8.9 months (95% CI 5–12.7); *p* = 0.127]. However, a significant difference emerged between patients with 20–29% and 30–50% blasts (*p* = 0.006). 

In addition, the type of response after the 4th cycle significantly influenced OS [CR mOS 19.5 months vs. PR 15.3 months vs. SD 8.9 months vs. progression of the disease (PD) 5.7 months, *p* = 0.011] (Figure 4), with no differences between CR and PR (*p* = 0.449). However, significant differences were observed comparing CR vs. SD (*p* = 0.008) and PR vs. SD (*p* = 0.043) after four cycles.

The achievement of CR or PR as the best response during HMA treatment was significantly associated with better outcomes in terms of mOS (*p* = 0.001). Indeed, no differences were seen comparing CR vs. PR (*p* = 0.174) subgroups, while significant differences were observed comparing PR vs. SD (*p* = 0.15) and CR vs. SD (*p* < 0.001).

Patients at adverse risk according to the ELN2017 classification had the worst survival (mOS 4.4 months vs. favorable risk 19.5 months, *p* < 0.001; vs. intermediate risk 10.7 months; *p* = 0.003). 

The presence of complex karyotype (CK) [3.3 months (95% CI 1.3–5.3) vs. absence of CK [11.0 months (95% CI 9.8–14.3), *p* = 0.003] and a proportion of marrow blasts ≥ 30% after the 4th cycle of HMAs [9.1 months (95% CI 3.7–14.5) vs. <30% 16.2 months (95% CI 12.3–20.0), *p* = 0.034] were significantly associated with poor outcome. Notably, no difference was observed in terms of mOS between patients with de novo AML [9.0 (95% CI 6.8–15.1) and s-AML [6.8 months (95% CI 6.8–15); *p* = 0.206]. 

The gender (*p* = 0.752), renal function (*p* = 0.124), BMI (*p* = 0.120), monosomal karyotype (*p* = 0.415), infectious complications (*p* = 0.203) and transfusion requirements at diagnosis (*p* = 0.138), and the occurrence of treatment-related adverse events (AEs) (*p* = 0.699) did not significantly affect survival with HMA therapy. The Univariate Cox Proportional Hazards Models for OS are presented in Appendix A. 

In multivariate analysis, age ≥ 80, CCI ≥ 3, CK and the type of best response (≥PR) during treatment maintained independent significance in predicting survival (Table 3), with a trend toward statistical significance for bone marrow blast percentage at diagnosis.

### 3.4. Event-Free Survival

The median EFS (mEFS) was 7.3 months (95% CI 5.0–9.6), 1-year EFS was 33.6% (95% 27.1–40.1) and 2-years EFS was 16.3% (95% CI 11–21.6). 

All factors that significantly influenced mOS, also had an impact on mEFS, except for the number of blasts after the 4th HMA cycle (Table 2). In addition, s-AML patients showed inferior mEFS compared to de novo AML [4.2 months (95% CI 2.0–6.4) vs. 8.7 (95% CI 5.9–11.6), respectively, *p* = 0.005). Univariate Cox Proportional Hazards Models for EFS are reported in Appendix A. However, high bone marrow blasts (≥50%) at diagnosis [HR = 2.57 (95% CI 1.3.9–4.75); *p* = 0.003], carrying a complex karyotype [HR = 2.10 (95% CI 1.08–4.08); *p* = 0.029]; transfusion independence [HR = 0.50 (95% CI 0.25–0.84); *p* = 0.001] and obtaining a CR as best response [HR = 0.39 (95% CI 0.22–0.70); *p* = 0.001] were the only factors that significantly influenced mEFS in multivariate analysis. Interestingly, having an ELN2017 high-risk did not significantly affect mEFS [HR = 1.48 (95% CI 0.99–2.20); *p* = 0.053) in this specific analysis. 

### 3.5. Long-Lasting Treated Patients

In our cohort, 57 patients (44 in the AZA group and 13 in the DEC group) received a median of 18 (IQR 12–46) cycles of HMAs (long survivors). The median OS and EFS were 24.3 months (95% CI 20.2–28.3) and 23.2 months (95% CI 18.9–27.2), respectively.

In this subgroup, 36 (63.2%), 10 (17.5%) and 11 (19.3%) patients achieved CR, PR and SD, after 4 HMAs cycles, respectively. These patients showed a significantly lower bone marrow blasts count (<50%) (89% vs. 62%; *p* < 0.001), higher eGFR (>60 mL/min) (95% vs. 69%; *p* = 0.007), lower ECOG (0–1) (98% vs. 83%; *p* = 0.021) and CCI (<6) (65% vs. 46%; *p* = 0.021) at baseline and the majority of them were stratified as ELN fav-int risk (72% vs. 46%; *p* = 0.001) compared with the 163 patients who received < 12 cycles. The median time from diagnosis to the start of therapy did not significantly differ between long survivors (16 days; IQR 10–35) and others (20 days; IQR 11–32) (*p* = 0.851).

### 3.6. Toxicities

In our cohort, 109 patients (49.5%) died from AML progression, 49 (22.2%) from infectious complication and 20 (9%) from other causes. Among infectious complications, the most frequent event was radiologically documented pneumonia which occurred in 35 patients (11%), followed by septic shock which was lethal in 12 patients (5.4%).

Of the 126 evaluable patients, 122 had 1 or more AE: of these, 103 were infections, 8 hemorrhages, 9 other non-hematological toxicities, and 2 infections + hemorrhages. Thirty-two patients (14.5%) experienced 2 or more different infectious events during HMA treatment. The incidence of any grade toxicities was similar between the AZA and DEC groups (62% vs. 58%; *p* = 0.33). No statistically significant differences concerning the type of infection (*p* = 0.714), the grade of severity (*p* = 0.429), and the number of neutrophils at the onset of the event (*p* = 0.079) were found between the two HMAs. The all-cause 30-day mortality of the entire cohort was 6.3%, not significantly different between treatment groups, although an increasing trend for DEC (AZA 3.8%, DEC 8.9%; *p* = 0.111) was found. Excluding early deaths related to rapid progression of the disease, the 30-day mortality of the whole population studied was 3.9% (AZA 2.6%, DEC 6.8%, *p* = 0.159). 

Grade 3 and 4 infectious events were observed in 62 patients (28.1%). Among them, 42 patients (19%) developed radiologically documented pneumonia, 15 patients (6.8%) showed a documented septic shock and 4 patients (2.2%) experienced other infections. The occurrence of pneumonia was not correlated with age ≥80 years (*p* = 0.528), CCI ≥3 (*p* = 0.077) and ECOG ≥2 at diagnosis (*p* = 0.693). Three (1.3%) of the nine patients who had significant bleeding during HMAs died (2 patients with intracranial bleeding and 1 patient with massive gastrointestinal bleeding).

The median time between the start of treatment and the first complication was 60 days (IQR 24–153) and 65 days (IQR 20–189) for the AZA and DEC groups (*p* = 0.868), respectively. The median time between the start of treatment and the occurrence of a second or third complication was 177 days (IQR 123–522) and 360 days (IQR 266–680), respectively.

Data regarding the need for hospitalization of the whole population were not available, not allowing an observational analysis or calculation of the hospitalization rate.

## 4. Discussion

In this large retrospective Italian study, we investigated outcomes of very elderly patients treated with HMAs for AML. We focused on patients older than 75 years, an age, therefore, in which comorbidities magnify the importance of elevated frailty of these patients, as evidenced by the elevated CCI ≥ 3 in half of them. Many elderly patients with AML are not even referred to the attention of hematologists, given the perceived dismal prognosis. Indeed, a large proportion of hematologists consider such elderly patients with AML as candidates for best supportive care (BSC) only, as reported in a large cohort, where only 38.6% received leukemia therapy within three months of diagnosis [16]. Indeed, the main purpose of our study was to identify specific characteristics of elderly patients, and, to our knowledge, this is the first report to focus on baseline organ functions (e.g., eGFR, CCI, BMI) in AML patients aged >75 years.

Intensive induction strategies are rarely used for older patients in community oncology practice [17], with comorbidities being the major cause of contraindication; indeed, half of the patients had a CCI ≥ 3. HMAs have been the first drugs to improve the natural life expectancy of patients with a diagnosis of AML that is expected between 5 and 6.5 months, in pivotal studies [9,10]. However, in these two phase 3 randomized trials, azacitidine produced an mOS of 10.4 months (*p* = 0.1) [10] and decitabine produced a median OS of 7.7 months that were not statistically different from the control arm [9]. Real-world data and real-world evidence have historically been utilized in the post-approval setting to assess the real-world application, efficacy, and safety of approved therapies [18]. In the setting of patients treated with HMAs for MDS at high risk, it emerged that in real life, approximately half of patients do not receive HMA therapy and in those who initiate it, approximately 35–45% receive less than four cycles and 41–69% received less than five or six cycles [19]. In our experience, 42.2% received less than four cycles, confirming the difficulties of maintaining drug exposure, probably for intrinsic resistance of some AML to HMAs, logistical problems and toxicity due to early infections. 

In the setting of patients with AML treated frontline with HMAs, many real-life studies have been published. The largest USA registry study, using the Surveillance, Epidemiology, and End Results-Medicare linked database, identified 2263 patients, of which 51% received azacitidine and 49% decitabine. Median OS from diagnosis was 7.1 and 8.2 months (*p* = 0.01) for azacitidine- and decitabine-treated patients, respectively. It is important to note that only 41.8% received more than four cycles of drugs and that only 34.3% followed a 7-day regimen of azacitidine [20]. The largest European registries study identified 1199 patients (age > 70 years) treated with intensive chemotherapy and 1073 with HMAs. In this study, CR and CRi were 56.1% and 19.7% with chemotherapy and hypomethylating agents, respectively, and this translated into similar mOSs of 10.9 (95% CI: 9.7–11.6) and 9.2 months [21], respectively. 

An Italian study in 306 elderly patients with AML treated with DEC showed an ORR of 48.4%. Seventy-one patients (23.2%) had CR, and thirty-two (10.5%) PR; median OS was 11.6 months for patients with favorable-intermediate cytogenetic risk and 7.9 months for those with adverse cytogenetic risk [22]. However, another Italian study reporting 78 patients treated with AZA and 32 with DEC as first-line therapy showed no differences according to cytogenetic status [23]. Our mOs of 8.2 months is therefore in line with other experiences and seems only minimally influenced by the more advanced age of our cohort of patients. Unfortunately, the results of HMAs in the treatment of unfit patients with AML are not satisfactory, and this is evident also for advanced ages. The main cause of death in our study remains AML progression and in the second instance, infections that may be in part due to immunosuppression from HMA and AML persistence. The most promising option we have at the moment for elderly patients with AML is the addition of anti-BCL-2 Venetoclax (Ven) to HMAs, as recently reported in the VIALE-A study comparing Ven/AZA vs. placebo/AZA. The Ven experimental arm (median age 76 years) induced a CR/Cri rate of 66.4%, mOs was 14.7 months and two-year survival was 35%, which represented a breakthrough in the field of low-intensity therapies [12]. Unfortunately, the first two real-life studies of HMA+Ven showed an mOS of 12.3 months [24] and 12.7 months [25], considerably shorter than the pivotal trial. Results from our study pose two different interpretations. The first is that HMAs only minimally improve the outcome of elderly patients with respect to BSC, CR/CRi are limited and therefore it is advised to add venetoclax whenever possible. The second consideration is that certain patients with favorable characteristics can reach prolonged responses to HMAs, sparing the hematological toxicity frequently seen with the adjunct of venetoclax. Probably more data will be needed to answer this dilemma, in the meanwhile clinical judgment is advised when considering the initiation of treatment. A limit of our study is that patients with low-ELN risk AML, potential candidates for HMAs only, are 8.3% of the total, and therefore not well represented in this age group. Our data, even if not specifically focused on this comparison, found AZA and DEC to be particularly similar in this elderly patients population, both in terms of efficacy and safety, therefore the choice was based mainly on the logistic preference of the patient and/or caregiver (7 days vs. 5 days, subcutaneous vs. intravenous). The only significant difference is the higher blast counts in the DEC group which may reflect the different rules governing the possibility of prescribing the two drugs in Italy that lasted up to 2017 for DEC, while AZA was not reimbursed for AML with > 30% BM blasts.

In a meta-analysis that included nine studies conducted with DEC, subgroup analyses of age, cytogenetics risk, AML type and bone marrow blast percentage showed no significant differences in treatment response to decitabine [26]. Nevertheless, our study found transfusion independence, obtaining a CR as the best response, and the absence of complex karyotype to be associated with improved survival. Additionally in the PIAZA study, transfusion independence was reported in 64 of 89 patients and was associated with improved progression-free survival [27]. Similarly, in our series, patients with a higher rate of bone marrow blasts showed inferior chances of response to HMAs.

In a large set of real-world data from the German registry, there is new evidence that the delayed time from diagnosis to treatment (TDT) does not affect prognosis in the setting of fit patients’ candidate to intensive treatment [28]. Nowadays, this is reflected in the clinical approach of waiting for molecular and cytogenetic characterization of AML to better select the intensive approach and target drugs available. Nevertheless, we assessed for the first time whether TDT could affect outcomes of very elderly patients treated with HMAs. Stratifying TDT in three different time intervals (<15 vs. >15–30 vs. >30 days), we found no significant difference in OS. Furthermore, we also reported that TDT is not different in long-term survivors than short-term in patients receiving HMAs. It is interesting to note that the length of TDT (more than 15 days) indicates that elderly patients are probably screened carefully for organ functions (e.g., echocardiography, spirometry, gait tests) and hematologists may attend safely molecular testing for factors predictive of response to HMAs (e.g., complex karyotype). In the near future, molecular definition will be more relevant to establish the best therapy; for example, IDH1-mutated AML could receive oral ivosidenib and subcutaneous or intravenous azacitidine, since in a recent phase III trial this treatment showed an impressive median OS of 24.0 months [29].

The most frequent serious AEs occurring with similar frequency in the azacitidine arm included febrile neutropenia (25.0%), pneumonia (20.3%), and pyrexia (10.6%) [10], while in the trial of decitabine the most common serious AEs were febrile neutropenia (24%), pneumonia (20%) and PD (11%) [9]. In our study, AEs were similar to the pivotal trials; therefore, also in a population characterized by more advanced age and several comorbidities, HMAs can be considered a feasible regimen in patients with AML.

Previous reports indicated an increase in toxicity in patients with renal insufficiency [30,31]; in our study, having CKD was more frequent in advanced ages and associated with frailty [32], and having reduced kidney function (GFR < 60) was associated with the worst outcome.

## 5. Conclusions

In conclusion, AML in the setting of very advanced age continues to pose a significant challenge in the treatment. Future molecules such as the hedgehog signaling pathway inhibitor, glasdegib, two IDH inhibitors, ivosidenib (IDH1 inhibitor) and enasidenib (IDH2 inhibitor), the fully absorbable oral formulation of the hypomethylating agent decitabine and partially orally absorbable azacitidine will be, together with venetoclax, the expanding armamentarium against AML in the elderly [33].

## Figures and Tables

**Figure 1 cancers-14-04897-f001:**
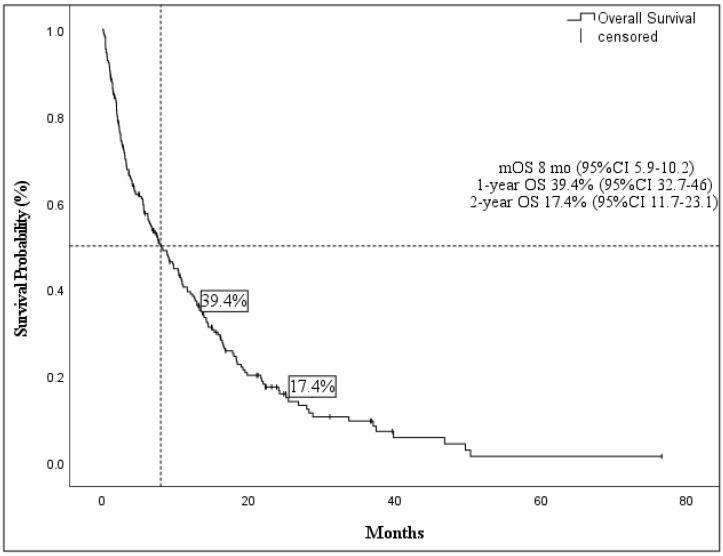
Median overall survival and 1- and 2-year overall survival of the entire cohort.

**Figure 2 cancers-14-04897-f002:**
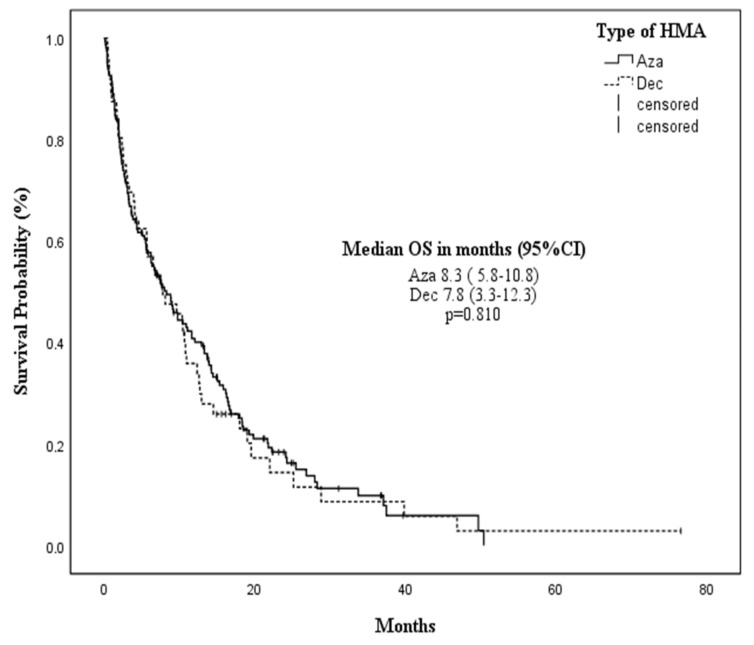
Median overall survival according to the hypomethylating agent.

**Figure 3 cancers-14-04897-f003:**
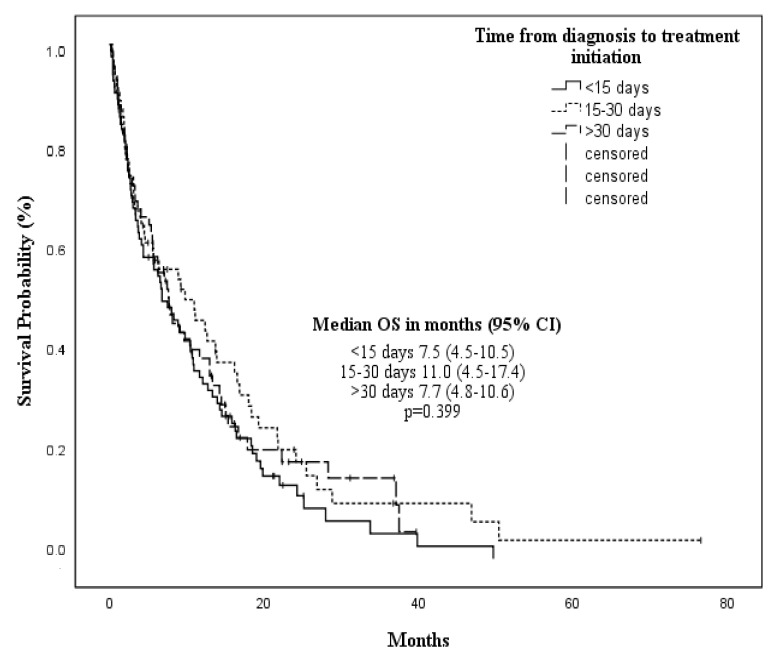
Median overall survival according to time from diagnosis to treatment initiation (≤15 days, 15–30 days, ≥30 days).

**Figure 4 cancers-14-04897-f004:**
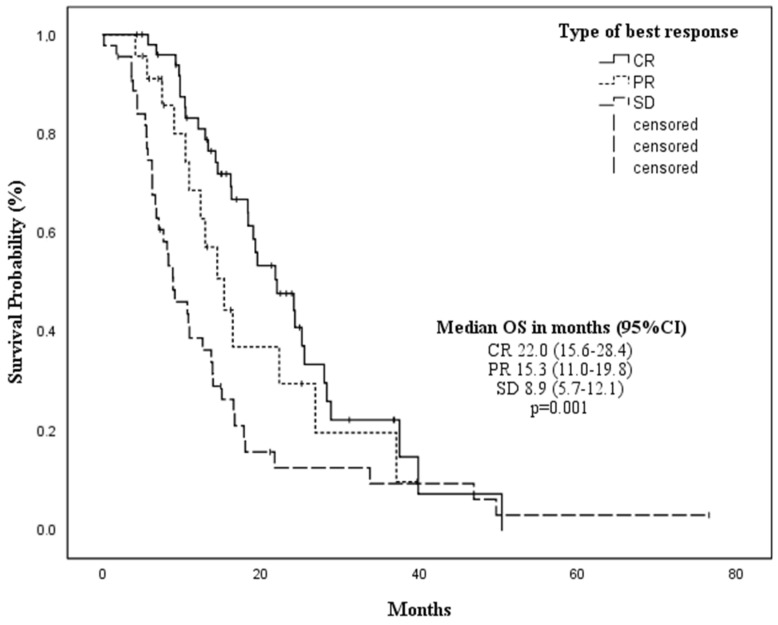
Median overall survival according to the type of response after four cycles of treatment.

**Table 1 cancers-14-04897-t001:** Baseline characteristics of patients.

Characteristics	No. of Patients = 220
**Male, *n* (%)**	119 (54)
**Female, *n* (%)**	101 (46)
**Median age at diagnosis (IQR)**	78.2 (75–86.2)
**Hb g/dL (median, IQR)**	8.8 (7.9–10.0)
**WBC × 10^9^/L (median, IQR)**	3.24 (1.6–9.8)
**Platelet count × 10^9^/L (median, range)**	56 (32–92.5)
**BMI, *n* (%)**	
<25 ≥25	97 (44.1) 123 (55.9)
**ECOG PS, *n* (%)**	
<2 ≥2	132 (60) 88 (40)
**eGFR, *n* (%)**	
<60 mL/min ≥60 mL/min	37 (16.8) 183 (83.2)
**BM blast percentage, *n* (%)**	
20–30% 30–50% >50%	78 (35.5) 74 (33.6) 68 (30.9)
**CCI, *n* (%)**	
<3 ≥3	44 (20) 176 (80)
**AML type, *n* (%)**	
De novo-AML s-AML	135 (61.4) 85 (38.6)
**ELN risk stratification,** **data available *n* (%)**	205 (93.1)
Favorable Intermediate Poor/adverse	17 (8.3) 100 (48.8) 88 (42.9)
**Infection at diagnosis, *n* (%)**	
No Yes	171 (77.7) 49 (22.3)
**Transfusion requirement, *n* (%)**	
No Yes	83 (37.7) 137 (62.3)
**Median No of cycles (IQR)**	5 (2–12)
**TDT, *n* (%)** <15 days 15–30 days >30 days	85 (38.6) 64 (29.1) 60 (32.3)

Abbreviations: AML, acute myeloid leukemia; BM, bone marrow; CCI, Charlson comorbidity index; ECOG PS, Eastern Cooperative Oncology Group Performance Status, eGFR, estimated glomerular filtration rate; ELN, European Leukemia Net; Hb, hemoglobin; s-AML, secondary AML; TDT, time from diagnosis to the start of therapy; WBC, white blood cell.

**Table 2 cancers-14-04897-t002:** Results of Kaplan–Meier overall survival and event-free survival analysis.

Variable	Median OS, Months (95% CI)	*p* Value *	Median EFS, Months (95% CI)	*p* Value *
**Age, years**				
<80	10.5 (7.4–13.5)	0.001	8.7 (5.0–9.6)	0.002
≥80	6.2 (3.0–9.5)		4.0 ( 2.1–5.8)	
**Gender**				
Male	7.1 (4.0–10.1)	0.752	5.6 (2.1–9.1)	0.552
Female	9.0 (6.3–11.8)		7.8 (5.1–10.5)	
**ECOG**				
0–1	9.8 (7.5–12.0)	<0.001	10.9 (7.5–14.3)	0.051
≥2	2.3 (1.3–3.8)		5.7 (3.5–7.9)	
**BM blast count at diagnosis**				
20–29%	13.7 (9.8–17.6)	<0.001	12.1 (8.7–15.5)	<0.001
30–50%	8.9 (5.0–12.7)		6.2 (2.6–9.7)	
>50%	4.1 (1.7–6.5)		3.7 (2.2–5.2)	
**Type of AML**				
de novo AML	9.0 (6.5–11.5)	0.206	8.7 (5.9–11.6)	0.005
s-AML	6.8 (3.3–10.3)		4.2 (2.0–6.4)	
**Infectious at AML diagnosis**				
No	9.7 (7.3–12.1)	0.203	8.7 (6.5–10.9)	0.126
Yes	4.6 (1.7–8.0)		3.4 (1.9–4.8)	
**eGFR (ml/min/1.73 m^2^)**				
≥60	9.2 (6.9–11.4)	0.124	8.6 (6.7–10.5)	0.196
<60	3.1 (1–5.3)		2.6 (1.5–3.7)	
**BMI at diagnosis**				
<25	10.7 (7.4–14.0)	0.120	8.6 (5.9–11.3)	0.630
≥25	6.2 (5.1–7.4)		5.7 (3.6–7.7)	
**CCI**				
<3	14.3 (7.9–20.7)	0.029	13.3 (8.0–18.7)	0.007
≥3	6.7 (5.1–8.2)		5.2 (3.0–7.4)	
**Transfusion dependency at diagnosis**				
No	12.1 (9.8–14.3)	0.138	9.3 (7.1–11.6)	0.391
Yes	6.0 (3.0–9.0)		5.0 (2.1–7.8)	
**Complex karyotype**				
No	11.0 (6.8–15.1)	0.003	9.8 (7.8–11.8)	<0.001
Yes	3.3 (1.3–5.3)		2.1 (1.3–2.9)	
**ELN risk classification**				
Favorable	19.5 (8.1–31.0)	<0.001	19.4 (8.8–30.1)	<0.001
Intermediate	10.7 (6.5–14.8)		10.6 (7.5–13.7)	
Adverse	4.4 (1.7–7.0)		3.1 (1.6–4.4)	
**Time from diagnosis to treatment initiation**				
<15 days	7.5 (4.5–10.5)	0.399	5.8 (2.5–9.1)	0.211
15—30 days	11.0 (4.5–17.4)		9.8 (7.9–11.7)	
>30 days	7.7 (4.8–10.6)		5.7 (2.6–8.8)	
**Type of HMA**				
Aza	8.3 (5.8–10.8)	0.810	7.3 (4.5–10.1)	0.947
Dec	7.8 (3.3–12.3)		6.2 (1.7–10.6)	
**BM blast count after 4th cycle**				
<30%	16.2 (12.3–20.0)	0.034	14.3 (11.4–17.1)	0.069
≥30%	9.1 (3.7–14.5)		8.9 (4.6–13.1)	
**Response after 4th cycle**				
CR	19.5 (12.9–26.2)	0.011	17.5 (12.2–22.8)	0.026
PR	15.3 (11.6–19.1)		14.6 (10.8–18.4)	
SD	8.9 (6.0–11.7)		8.5 (4.6–12.3)	
**Best response**				
CR	22.0 (15.6–28.4)	0.001	19.5 (14.7–24.2)	<0.001
PR	15.3 (11.0–19.8)		13.2 (8.9–17.5)	
SD	8.9 (5.7–12.1)		8.7 (6.5–10.9)	
**Transfusion independence**				
Yes	19.3 (15.4–23.2)	<0.001	17.7 (14.0–21.3)	<0.001
No	3.3 (0.8–5.8)		4.2 (2.6–5.8)	
**Treatment-related complication**				
No	10.8 (0.6–27.3)	0.699	8.5 (6.1–10.9)	0.870
Yes	5.7 (3.7–7.6)		6.6 (2.6–10.6)	

Abbreviations: AML, acute myeloid leukemia; AZA, azacitidine; BM, bone marrow; CCI, Charlson comorbidity index, CR, complete remission; DEC, decitabine; eGFR, estimated glomerular filtration rate; ELN, European Leukemia Net; HMA, hypomethylating agents; OS, overall survival; PD, progression disease; PR, partial remission; SD, stable disease; s-AML, secondary AML; ** p* value significant < 0.05.

**Table 3 cancers-14-04897-t003:** Multivariate analysis for overall survival (Cox proportional hazards regression model).

Covariate	HR (95% CI)	*p* Value
**BM blast count at diagnosis** ≥50%	1.69 (0.99–2.90)	0.054
**Age at diagnosis** ≥80 years	2.26 (1.23–4.16)	0.009
**CCI** ≥3	1.97 (1.05–3.69)	0.034
**Presence of CK**	2.89 (1.51–5.49)	0.001
**Type of best response** ≥PR	0.22 (0.12–0.40)	<0.001

Abbreviations: BM, bone marrow; CI, confidence interval; CCI, Charlson comorbidity index; CK, complex karyotype; HR = hazard ratio; PR, partial remission.

## Data Availability

The original contributions presented in the study are included in the article/Appendix A. Further inquiries can be directed to the corresponding author.

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
