# Peer review of "Identification of Predictive Factors for Overall Survival and Response during Hypomethylating Treatment in Very Elderly (≥75 Years) Acute Myeloid Leukemia Patients: A Multicenter Real-Life Experience"

_cancers, 2022, doi:10.3390/cancers14194897_

Round 1
Reviewer 1 Report
The paper by Molica et al., addresses in a real life setting an important question on very old patients suffering from AML treated with HMAs with important data about biological features and ELN classification. This is, up to date, a big lack in the literature and adds values on the analysis performed.
Only one formal note: "5 + 2 + 2" schedule to be corrected with "5+2" in the paragraph patientd and methods
Author Response
thank you for your comments.
5+2+2 schedule has been corrected with 5+2.
Reviewer 2 Report
I have the following minor concerns in the very well presented and almost excellent discussion (i really liked it, almost covered everything, deep knowledge of the topic by the authors):
1. Usually when MDS patients receive the hypomethylating agent azacitidine there are two possibilities. Some do not respond from the beginning and some others respond amazingly for the first few months and then when they lose their response, the outcome is dismal. Please report that to the discussion, provide your own experience and report real-life data on this question.
2. What is the opinion of the authors regarding the addition of venetoclax to the hypomethylating agents for these elderly patients? What are the real-life data of the authors? Does venetoclax help or does it cause more toxicity, thereby causing problems instead of helping these frail elderly patients?
3. Please correct some few grammatical /syntactical errors through the whole manuscript. for example, just to cite a few: discussion:
[18], with co-morbidities BEING the major cause of contraindication;
indeed, in our cohort half of ......etc , please remove 'in our cohort'
and hematologist may attend safely molecular testing ........please correct 'hematologists may attend....etc.
Author Response
- The retrospective analysis included only very elederly AML patients treated frontline with HMAs. No patients with previous MDS had been previously treated with HMAs. Data collected specifically referred to >75 years AML patients, so we have no data on MDS patients.
- Results from our study pose two different interpretations. The first is that HMAs only minimally improve the outcome of elderly patients respect to BSC, CR/CRi are limited and therefore it is advised to add venetoclax whenever possible. The second consideration is that certain patients with favorable characteristics can reach prolonged response to HMAs, sparing the hematological toxicity frequently seen with the adjunct of venetoclax. To date, we have a low number of very elederly patients (>75 years) treated with HMAs + Ven and therefore we cannot describe this population in the study. However, we are noticing that the incidence of toxicities do not signficant differ in this setting comparing with a younger subgroup.
- We corrected the grammatical /syntactical errors.
Reviewer 3 Report
Minor comments:
(1) Some denotations in the main context are not correct. On page 5, it says “Results of univariate logistic regression model analysis for CR are provided in supplemental Table 2”, but supplemental Table 2 is “Univariate Cox Proportional Hazards Models for Overall Survival”.
(2) Also on page 5, the variables that are associated with higher CR odds are presented with OR, CI value and p-value. But the p value for CCI<3 is not seen here.
(3) Could you specify the test statistic (chi-square?) that was used for Log Rank Test in Kaplan-Meier overall survival results shown in Table 2? Why there is only one p-value for a variable with three categories e.g. “Time from diagnosis to treatment initiation”, “Response after 4th cycle”? How do you interpret the single p-value for these categorical variables with more than two categories?
(4) Could you highlight or circle the points where median, 1- and 2-year OS indicate on Figure 1? Because that will be more intuitive for readers.
(5) Are the grade 3 and 4 infectious events described on page 10 prior to or post AZA/DEC treatment? Although the infection status at diagnosis seems to have no significant impact on median OS and EFS, do you also do follow-up checks on how patients’ pre-treatment infection status (if any) changes over time after taking AZA/DEC?
Author Response
- We apologize for the inconvenience; results of univariate regression model analysis for CR are depicted in supplemental Table 1. We corrected the sentence on the text.
-
We corrected this mistake in the text adding p value (p<0.001) for CCI<3.
-
We used Log Rank (Chi Square test); we reported on the table only global p value but, as you properly suggested, we added p value resulted from multiple comparison of Kaplan-Meier curves on the manuscript. TDT was not significant between the thre groups, therefore we have reported only the global p-value.
- we created the new figure rquested.
-
The grade 3 and 4 infectious events described on page 10 were recorded during the whole treatment period.
-
We are grateful for your suggestion. Although the undoubted limits of the retrospective nature of the study, we made an attempt to answer your request evaluating whether the rate of infections differ based on pre-treatment infection status but we did not find any significant difference (43% vs 55%, p=0.147).
Reviewer 4 Report
This paper reports a real-world-data of elderly AML patients who were treated with HMAs.
Although the results of this paper by HMAs agree with my practical feelings, it lacks new information and therefore the paper’s impact is low. Comparing the results of patients who received intensive therapy other than HMA treatment, or who chose no treatment, it should be made clear what kind of patient group is suitable for HMA.
The treatment of choice is either AZA 75 mg/m(2) for 7 days or DEC 20 mg/m(2) for 5 days. Were there any patients treated with dose reductions? If there were, what criteria was used for them?
A limitation of this paper is that it analyzes only OS and EFS and does not refer to patient QOL. Analyzing data such as length of hospitalization improves the impact of papers.
Author Response
- We appreciate your suggestions. However, we cannot compare this very elderly AML subgroup treated whit HMAs with patients who recieved intensive chemotherapy (no patient of this age is a candidate for intensive chemotherapy in our center). In our analysys only 20 patients had been recieved BSC, therefore the comparison was not statistically applicable. In our opinion, the significant information is the fact that the analysis includes only very elderly patients (>75 years) and that in the literature there is currently no real life analysis that takes into consideration only this subgroup of patients (the median age of other experiences was between 68 and 73 years).
- We exluded form the analysys patients with HMAs dose reductions at baseline
- We did not assess QoL of these patients. The only data that emerged was that the average hospitalization time of patients who developed an infectious complication during treatment was 19 days (IQR 7-35 days).